# Plasmonic nanostar photocathodes for optically-controlled directional currents

Jacob Pettine [1,2], Priscilla Choo [3], Fabio Medeghini [1], Teri W. Odom [3,4✉] & David J. Nesbitt[1,2,5✉]

Plasmonic nanocathodes offer unique opportunities for optically driving, switching, and steering femtosecond photocurrents in nanoelectronic devices and pulsed electron sources. However, angular photocurrent distributions in nanoplasmonic systems remain poorly understood and are therefore difficult to anticipate and control. Here, we provide a direct momentum-space characterization of multiphoton photoemission from plasmonic gold nanostars and demonstrate all-optical control over these currents. Versatile angular control is achieved by selectively exciting different tips on single nanostars via laser frequency or linear polarization, thereby rotating the tip-aligned directional photoemission as observed with angle-resolved 2D velocity mapping and 3D reconstruction. Classical plasmonic field simulations combined with quantum photoemission theory elucidate the role of surface-mediated nonlinear excitation for plasmonic field enhancements highly concentrated at the sharp tips ($R_{tip} = 3.4$ nm). We thus establish a simple mechanism for femtosecond spatiotemporal current control in designer nanosystems.

[1] JILA, University of Colorado Boulder and National Institute of Standards and Technology, Boulder, CO 80309, USA. [2] Department of Physics, University of Colorado Boulder, Boulder, CO 80309, USA. [3] Department of Chemistry, Northwestern University, Evanston, IL 60208, USA. [4] Department of Materials Science and Engineering, Northwestern University, Evanston, IL 60208, USA. [5] Department of Chemistry, University of Colorado Boulder, Boulder, CO 80309, USA. ✉email: todom@northwestern.edu; djn@jila.colorado.edu

Femtosecond optical control over nanoscale currents is essential for ultrafast electron diffraction and microscopy[1–6], x-ray free-electron lasers[7–9], and terahertz optoelectronic circuits[10–12]. Among recent advances, plasmonic metal nanostructures have shown considerable versatility and promise as bright photocathodes[8,9,13], femtosecond photodiodes[10,11], and carrier-envelope-phase-sensitive photodetectors[14,15] that can be integrated into nanoscale, chip-based devices. In plasmonic systems, the mapping from optical field parameters onto near-field electron dynamics is primarily governed by the particle geometry and corresponding field enhancements, which can be crafted with high precision by synthetic or lithographic methods[11,13,16–18]. Particle geometry defines the surface field-enhanced hot-spot regions where conduction electrons build up during collective oscillations, become excited, and escape as photoemission or photovoltaic currents. Geometry and particle array patterning also govern the frequency response of plasmonic systems, which have been tailored for broadband photodetection[19], photocurrent polarity control[20], and selective multi-mode lasing with narrow spectral linewidths[21]. Spectral characteristics can even be controlled at the single-particle level for asymmetric particles that support multiple resonances, such as gold nanostars[16,22–24].

With nanostars and other multi-resonant particles, important opportunities for spatiotemporal photocurrent control emerge via frequency- and polarization-selective excitation of different plasmonic hot spots, which are often spatially separated and oriented in different directions[16,17,22,24,25]. Electric near-field hot spots have been extensively investigated in nanoplasmonic systems, with photoemission electron microscopy (PEEM) studies establishing the correlation between photoemission and plasmonic hot spots with ~20 nm spatial resolution[16,17]. Furthermore, these techniques have been combined with optical pulse shaping[26] to achieve coherent control over spatial photoemission distributions on femtosecond timescales[18]. However, direct observation of the corresponding photoelectron momentum-space distributions has remained a challenge, requiring angle-resolved photoelectron mapping from single, resonantly excited nanoparticles. Such capabilities have only appeared recently[25,27–29] as a comprehensive understanding of photocurrent distributions is becoming crucial for the design and implementation of nanocathodes in nascent ultrafast nanoelectronics and electron imaging applications. Full photoelectron momentum and energy characterization has been achieved by Lehr et al.[28,29] on individual gold nanorods and bow-tie nanoantennas using time-of-flight momentum PEEM[28,29], which serves to clarify nanoplasmonic angular photoemission distributions and phenomena such as the transition into the optical field emission regime. Angular photoelectron mapping and steering have also been demonstrated for gold[30] and tungsten[31,32] nanotips, primarily in the field emission regime. Despite these advances, many important aspects of the nanoplasmonic photoelectron emission mechanism and opportunities for angular control remain to be elucidated, particularly in the multiphoton regime.

Here, we demonstrate optically-controlled directional multiphoton photoemission (MPPE) from single gold nanostars, using two-dimensional (2D) photoelectron velocity mapping and 3D reconstruction for detailed characterization of the angular and energy distributions. We begin with an examination of the plasmonic properties and directional photoemission from single-nanostar tips resonantly excited using a pulsed femtosecond laser. Individual tips behave as locally bright, point-like electron sources with a high degree of spatial coherence. We then demonstrate the selective excitation of different tips on a single nanostar via optical frequency and polarization control, yielding wide angular switching/steering of the tip-aligned photoemission currents. Correlated scanning electron microscopy (SEM) and 3D

photoemission modeling clarify the effects of nanostar tip geometry and plasmonic field enhancements. As the overall directional effects are not contingent on laser intensity, the present method for optically controlling photocurrents can be extended from the weak-field (multiphoton) regime into the strong-field (optical field emission) regime. However, weak-field MPPE processes are emphasized here due to the minimal nanostar heating, sub-single-electron femtosecond pulses ($10^{-5}$ photoelectrons from each pulse on average) that preclude space-charge effects, and <1 eV photoelectron kinetic energy spreads for high temporal coherence. Photocurrent control timescales approaching the attosecond range may be achievable, fundamentally limited only by the nonlinear photoemission decay associated with plasmon dephasing.

## Results

**Single-tip excitation and photoemission properties.** Gold nanostars are synthesized with sharp $3.4 \pm 0.4$ nm radii tips and sorted by size (see Methods) to select for simple geometries with an average number of three tips lying in the surface plane (Supplementary Fig. 1). Electron micrographs for a selection of representative nanostars are shown in Fig. 1a. Plasmonic properties are readily characterized using normal-incidence laser light for these primarily in-plane geometries, with the nanostars supporting localized surface plasmon resonances across the near-infrared tuning range of the Ti:sapphire oscillator (700–1000 nm, 50 fs, 75 MHz). A scanning sample stage enables diffraction-limited photoemission mapping for locating single nanostars dispersed on a transparent, conductive indium tin oxide (ITO)-coated glass substrate[27,33] (see Methods). We then collect photoelectron velocity distributions as a function of laser frequency, polarization, and intensity with the velocity map imaging (VMI) electrostatic lens configuration depicted in Fig. 1b[34].

We characterized the resonance properties for three sample nanostars via spectral (Fig. 1c) and polarization dependences (Fig. 1d), with the peak polarization indicated in the correlated SEM insets aligned along a particular tip for each nanostar. In each case, the $n$-photon photoemission ($n$PPE) rate varies as $\Gamma_{n\text{PPE}} \sim \cos^{2n}(\theta - \theta_{\text{tip}})$ as the polarization angle $\theta$ is rotated away from the resonant tip, consistent with the field projection onto a single, well-defined hot-spot axis. The essential role of the plasmon resonance in driving these behaviors has been established in previous photoemission studies correlated with dark field microscopy for gold nanorods[33], confirming the direct relation between surface plasmon (Mie) resonances in linear scattering spectra and the peaks observed in MPPE spectra. Due to the $E^{2n} \propto I^n$ electric near-field sensitivity of the $n$-photon process, nanostars with multiple similar tips may have only one dominant hot spot (e.g., Star 2). Note, for instance, that a mere 10% difference in field enhancement results in a factor of two difference in the 4PPE rate.

Electrons must absorb multiple visible or near-infrared photons (~1.5 eV) to overcome typical metal work functions (~5 eV) and escape into the vacuum. The total MPPE rate is described by a sum over process orders,

$$\Gamma_{\text{MPPE}} = \sum_n \Gamma_{n\text{PPE}} = \sum_n \sigma_n(\omega, \theta) I_0^n, \qquad (1)$$

in which $I_0$ is the input laser intensity and $\sigma_n(\omega, \theta)$ is the $n$PPE cross-section as a function of laser frequency and polarization. The plasmonic field enhancement effect is therefore included within $\sigma_n(\omega, \theta)$. We determined multiphoton process orders for a random selection of resonantly excited nanostars by measuring photoemission rate as a function of peak laser pulse intensity. The

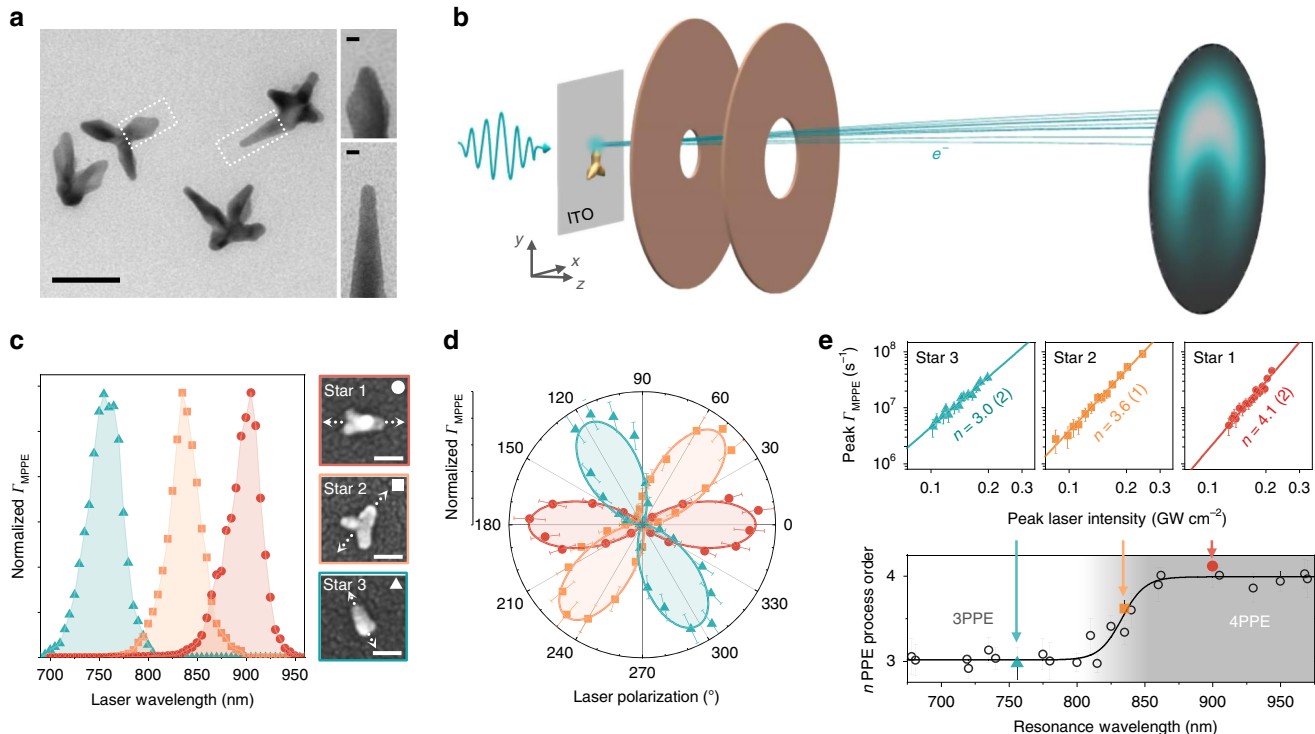

**Fig. 1 Single-nanostar plasmon resonance excitation and photoelectron velocity mapping. a** Transmission electron micrographs of representative few-arm nanostars (50 nm scale bar) with insets displaying individual tips (5 nm scale bars). **b** Experimental configuration, in which photoelectrons with initial transverse $(v_x, v_y)$ velocity are linearly mapped onto $(x,y)$ position on a spatially-resolved microchannel plate detector. **c** Three example nanostars, each with a single dominant plasmon resonance mode excited via polarization aligned along the arm axes, as indicated by the arrows (50 nm scale bars). **d** Polarization-dependent photoemission rate at peak resonance wavelengths for each of the three stars, characterizing the peak polarization settings. Data is fit to $\cos^{2n}(\theta - \theta_{\text{tip}})$ for multiphoton process order $n$. **e** Summary of multiphoton process orders for plasmon resonance modes on different nanostars, determined via linear intensity-dependence fits on log–log plots (power-law behavior) as demonstrated for the three example stars. The resulting process order data is fit to a sigmoidal curve from $n = 3.01(3)$ to $n = 3.99(4)$, demonstrating the clear transition from the three-photon to the four-photon regime with decreasing photon energy. Intermediate values between 3 and 4 are effective process orders representing a weighted average of three- and four-photon contributions. All reported parenthetical values are standard errors of the variance-weighted nonlinear least-squares fits.

process order summary in Fig. 1e reveals a sigmoidal transition from 3PPE to 4PPE centered around 830 nm (1.5 eV), indicating a nanostar tip work function of $\phi \approx 3\hbar\omega_{830} = 4.5$ eV at which three-photon-excited electrons from the Fermi level begin to overcome the surface potential barrier. Intensity-dependence plots and power-law fits are shown for the three sample nanostars, with the non-integer Star 2 process order ($n = 3.6$) representing weighted contributions from both 3PPE and 4PPE processes. From Eq. (1), the log–log slope is

$$\frac{d\log_{10}(\Gamma_{\text{MPPE}})}{d\log_{10}(I_0)} = \frac{\sum_n \sigma_n(\omega, \theta) I_0^n n}{\sum_n \sigma_n(\omega, \theta) I_0^n} = \sum_n w_n n, \quad (2)$$

which is the weighted average over process orders with weight factors $w_n = \sigma_n(\omega, \theta) I_0^n / \sum_{\tilde{n}} \sigma_{\tilde{n}}(\omega, \theta) I_0^{\tilde{n}}$ representing the relative contributions of each process order, such that $\sum_n w_n = 1$. With only 3PPE and 4PPE processes contributing, Eq. (2) yields an effective process order of 3.5 when $\sigma_3(\omega, \theta) I_0^3 = \sigma_4(\omega, \theta) I_0^4$, as is the case at 830 nm in Fig. 1e. It should be noted that peak photocurrents of $10^8$ s$^{-1}$ during the peak laser pulse intensity correspond to an average of only $10^{-5}$ photoelectrons from each pulse when multiplied by the laser pulse duration ($10^{-13}$ s). With a laser repetition rate of 75 MHz, we can thus achieve photoemission rates $>100$ s$^{-1}$ with negligible probability of two electron emission events occurring within a single laser pulse, thereby precluding any space-charge effects.

These measurements verify that all signal is predominantly due to MPPE rather than optical field emission or thermionic emission for the range of intensities utilized in these studies. For further verification, the Keldysh parameter, $\gamma = \sqrt{\phi/2U_p}$, is commonly used to characterize the transition from weak- to strong-field emission[35], where $\phi$ is the work function and $U_p = e^2 E^2/(4m_e\omega^2)$ is the ponderomotive energy. Perturbative MPPE is dominant for $\gamma > 2$ and optical field emission is dominant for $\gamma < 1$, with the transition occurring in the $1 < \gamma < 2$ range[9]. For peak input pulse intensities $I_0 < 3 \times 10^8$ W cm$^{-2}$ and simulated field enhancements $|E/E_0| < 100$ described below, Keldysh parameters $\gamma > 3$ fall within the MPPE regime and corroborate the measured intensity dependences.

**Single-tip photoelectron velocity distributions.** Selecting Star 1 (Figs. 1c and 2a) as a representative single-tip emitter, we measured and modeled the characteristic photoelectron velocity distribution from a resonantly-excited nanostar tip. First, we performed finite element simulations using the nanostar geometry measured via correlated SEM for insight into the plasmonic field enhancements. After we account for a 3 nm Pt coating applied prior to SEM imaging for improved contrast (see Methods), the nanostar arm angles, lengths, and widths in our models are determined from the correlated SEM micrographs, with the arm shapes and tip radii (3.4 nm) obtained by statistical transmission electron microscopy (TEM) characterization (Supplementary Fig. 1). Both the influence of the ITO-coated substrate and the HEPES (4-(2-hydroxyethyl)-1-piperazineethanesulfonic

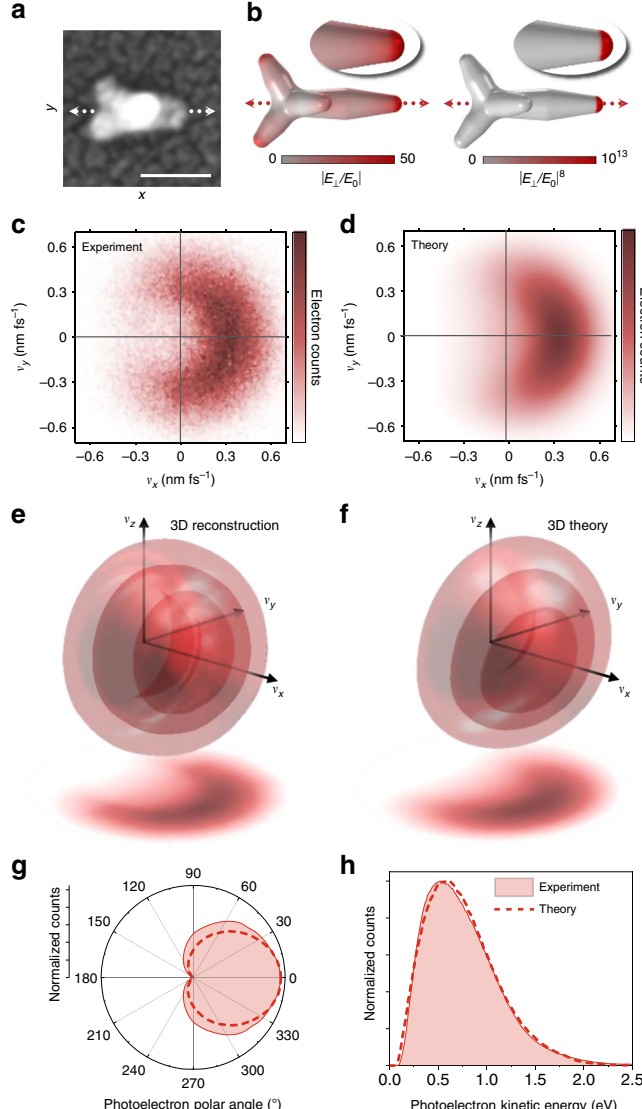

**Fig. 2 Characterization of single-tip multiphoton photoemission.**
**a** Scanning electron micrograph of a representative single-tip emitter (Star 1). The tips appear broader than in transmission electron micrographs and nanostar statistics due to a 3 nm Pt film deposited prior to electron imaging to improve sample conductivity and image quality. This added thickness is therefore subtracted from the measured nanostar dimensions for modeling and simulation. Scale bar: 50 nm. **b** Modeled nanostar geometry with calculated linear and eighth-order (four-photon) surface-normal electric field enhancements. All measurements and calculations are performed on-resonance at 900 nm and 0° polarization. The ITO-coated substrate and HEPES ligands (not shown) are included in the field calculations. **c** Experimental and **d** theoretical velocity map images of the $v_z$-projected photoelectron distribution. **e** Reconstructed experimental and **f** theoretical 3D velocity distributions, showing iso-probability surfaces at 75%, 50%, and 25% maximum along with the overall $v_z$ projections. **g** Angular distributions and **h** electron kinetic energy distributions determined from $xy$ slices of the 3D experimental and theoretical distributions.

acid) ligand layer on the nanostar surface were accounted for in the simulations. Further details of the finite element modeling are shown in Supplementary Fig. 2 and discussed in Supplementary Note 1. The simulated surface-normal field distribution in Fig. 2b highlights all of the tips in the linear case, with a particularly

strong peak enhancement of $|E_\perp/E_0| \approx 50$ at the resonantly-excited tip. This tip becomes clearly dominant when the field enhancement is raised to the eighth power for the relevant 4PPE process, with the nonlinear enhancement strongly confined to its apex. Photoemission distributions measured for this hemi-spherical hot spot geometry may also guide expectations for convex hot spots in other systems, including etched nanotip photocathodes[4,6].

The measured 2D ($v_z$-projected) photoelectron velocity map in Fig. 2c shows clear directionality along the resonant tip axis, as would be predicted for electrons emitted outward from the field-enhanced surface region. The 3D velocity distribution (Fig. 2e) was reconstructed using the basis set expansion (BASEX) algorithm[36], which relies on the approximation of cylindrical symmetry around the resonant $x$-axis tip to compensate for information lost in the $v_z$ projection (Supplementary Note 2). Tip-aligned directional emission is also observed for Stars 2 and 3 (Supplementary Fig. 3), which are resonantly excited in the transition regime and the 3PPE regime, respectively. While both 3PPE and 4PPE processes lead to similarly tip-aligned photoemission distributions, photon energies just above the 3PPE onset may be preferred to reduce the kinetic energy spread (Supplementary Fig. 3) for optimal temporal coherence of the photoelectron waveform[1].

Photoemission generally proceeds by a combination of volume and surface mechanisms for momentum conservation[37,38]. The directional distributions observed here indicate the dominant role of surface scattering, in which case the multiphoton excitation rate depends on the surface-normal electric field[37–40], $E_\perp^{2n}$. In contrast, volume excitation mechanisms depend on internal fields as $E^{2n}$ (Supplementary Fig. 2) and produce hot electrons throughout the nanostar that would escape from various surfaces around the tips and body, leading to largely isotropic rather than directional velocity distributions. We therefore restricted our present theoretical investigations to surface-mediated MPPE, which is further supported by the tip-localized photoemission observed in nanostar PEEM studies[16].

Calculations based on the coherent surface MPPE theory developed by Yalunin et al.[40] were carried out to determine photoemission rates and velocity distributions from each nanostar surface area element, using the full SEM-correlated nanostar geometry and simulated plasmonic fields shown in Fig. 2b, with a fine surface mesh shown in Supplementary Fig. 2. The full 3D photoelectron velocity distribution was then determined by summing all photoemission contributions over the entire nanostar surface (see Methods). An initial Fermi–Dirac electron distribution within the gold was considered at a pulse-averaged temperature of 2500 K, calculated via the two-temperature model (Supplementary Fig. 6 and Supplementary Note 4) with a linear nanostar absorption cross-section of $10^4$ nm$^2$ determined via the finite element simulations. The emission process at each nanostar surface area element consists of a heated Fermi gas impinging on the surface barrier and becoming excited via coherent multiphoton absorption into external Volkov (field-dressed) states. The outgoing photocurrent includes direct and surface-rescattered quantum amplitudes[40]. Optical field penetration into the gold is neglected and all excitation therefore occurs from the external evanescent component of the initial free-electron wavefunctions, which accounts for momentum non-conservation at the surface barrier. Due to the short decay length of the external electron wavefunction (~2 Å) and the small quiver amplitude of the emitted electrons (~1.5 Å) relative to the plasmonic field decay length (~1.5 nm), the external field is approximated to be uniform in calculating the emission amplitude. The evanescent plasmonic field then ponderomotively accelerates the emitted

electrons outward from the surface, uniformly shifting the kinetic energy distribution by $U_p$ as described further in the Methods. The ponderomotive energy $U_p = E^2 e^2/(4m_e\omega^2) = I_0 |E/E_0|^2 e^2/(2c\varepsilon_0 m_e\omega^2)$ for the present input intensity ($I_0 = 2 \times 10^8$ W cm$^{-2}$) and calculated tip field enhancement ($|E/E_0| = 50$) is only 0.03 eV. Although only a minor contribution in the present studies, the ponderomotive energy scales linearly with the input intensity and quadratically with the field enhancement and thus may become important in other similar systems.

Theoretical 2D (Fig. 2d) and 3D (Fig. 2f) distributions reiterate and confirm the tip-aligned directionality, with good agreement in the photoelectron angular (Fig. 2g) and kinetic energy (Fig. 2h) distributions. The slight downward directionality in the theoretical 3D distribution is due to the ITO image charge field, which skews the tip field enhancement toward the substrate (also see Supplementary Fig. 4). However, the otherwise strong agreement between experimental and theoretical distributions indicates that such substrate symmetry-breaking effects are minor. With regard to substrate effects, we also mention the possibility of gap resonances occurring in nanoscale junctions between nanoparticles and conducting substrates separated by an organic molecular layer such as the ~1 nm HEPES ligand layer on the nanostars, as observed previously with Au nanospheres separated from an Au film substrate by an organic spacer layer[41]. While no direct evidence of a gap resonance effect is observed in the present experiments nor in the finite element simulations (Supplementary Fig. 2), due primarily to the in-plane resonance excitation with in-plane polarized light, such effects can be difficult to accurately account for in the simulations as they depend sensitively on the details of the nanostar–substrate interface, which we do not directly observe.

Although we have investigated the near-field contributions to photoemission dynamics and distributions in detail in order to gain general insights for arbitrary emitter geometries, the nanostar tips studied here are among the sharpest nanoplasmonic geometries available and for most purposes can be treated as point-like electron sources. Electrons may therefore be emitted into a broad angular distribution (Fig. 2g) but still retain a high degree of spatial coherence, determined by the small source radius (3.4 nm) and photoelectron angular uncertainty (~70°). Nanostar tip spatial coherence characteristics are comparable to those of state-of-the-art femtosecond electron sources operating in the linear photoemission regime[42] (Supplementary Note 3), which are only an order of magnitude from the fundamental limit imposed by the uncertainty principle. Furthermore, single-nanostar tips remain stable emission sources in the present intensity range during hours of measurements, including at least half an hour of continuous exposure while collecting velocity maps, as demonstrated in Supplementary Fig. 5. We calculate only modest lattice temperature increase of 200 K during a laser pulse (Supplementary Figs. 6, 7 and Supplementary Note 4) with 13 ns between pulses sufficient for equilibration back to room temperature. The high degree of stability and spatial coherence enables ultrafast point projection[3,4,6] and diffractive[2,3,5] imaging of nearby nanoscale objects following possible electron beam manipulation such as acceleration or collimation via nanoengineered electron optics[7]. For higher optical intensities, a variety of new physical behaviors emerge in the strong fields and gradients at point-like nanotips[43], including the onset of tunneling emission and subsequent quiver or sub-cycle dynamics[44]. Such effects are negligible in the present MPPE studies, but the theory readily extends into intermediate- and strong-field regimes[40] and the influence of strongly varying plasmonic fields on the photoelectron trajectories can be included in the manner demonstrated by Dombi et al.[13] when necessary.

**Selective excitation of multiple nanostar hot spots.** Building on previous investigations of selective nanostar tip excitation[16], the excitation of in-plane tips with in-plane polarization control in the present studies provides a particularly clear mapping between optical parameters and tip hot spots. While the presence of many tips and the effect of near-field tip–tip coupling[22] can lead to complicated optical parameter mappings in some cases, we emphasize the simple, typical nanostar behaviors here and proceed to demonstrate selective tip excitation by independently tuning frequency and polarization. Multiple tips are involved in both plasmon resonance modes for the four-tip nanostar in Fig. 3a, but simulations in Fig. 3b reveal that only one tip hot spot is dominant for each mode. Spectra and polarization plots show two distinct plasmon peaks at nearly orthogonal peak polarization angles, with the photoemission rate at either peak showing minimal contributions from the other. Entirely frequency-controlled tip selectivity can thus be achieved for an isotropic polarization state (circular or unpolarized) or for a linear polarization state oriented between the two resonance modes. The frequency sensitivity depends on the spectral peak widths, relative amplitudes, and separation.

An intermediate laser frequency setting exists between the spectral peaks at which both plasmon modes are equally excited at their respective linear polarization angles. This is demonstrated by the four-lobed polarization dependence measured between the resonances at 835 nm in Fig. 3a and reiterated by the calculations in Fig. 3b for the correlated nanostar geometry. At the intermediate frequency, the linear polarization angle may thus be tuned to select between the two resonance modes and corresponding tip hot spots, demonstrating polarization-controlled selective tip excitation. The relative mode strength (i.e., relative lobe amplitude in the polarization plots) can be continuously adjusted via frequency tuning. Overall, the 2D optical parameter space sampled in Fig. 3 is described by the $n$PPE photoemission cross-section, $\sigma_n(\omega, \theta) = A_1^{(n)}(\omega)\cos^{2n}(\theta - \theta_1) + A_2^{(n)}(\omega)\cos^{2n}(\theta - \theta_2)$, in which $A_1^{(n)}(\omega)$ and $A_2^{(n)}(\omega)$ are the $n$-photon spectra of the two plasmon resonance modes. In addition to the strong theoretical agreement with the observed spectral and polarization behaviors (Fig. 3b), MPPE rates are calculated to within an order of magnitude of the experimental measurements by integrating the theoretical current density, $J_{MPPE}$, over the nanostar surface, accounting for both 3PPE and 4PPE contributions. Reserving other aspects of the theory, this level of quantitative agreement corresponds to calculated fields within 50% of the experimental values for a 3PPE process ($\propto E^6$), which is good considering the sensitivity of the field enhancements to the precise tip radius, the charge distribution during plasmon oscillation (i.e., the overall nanostar geometry), and the surface dielectric environment due to the HEPES surface ligands (see Methods and Supplementary Note 1). Note that while the spectra of the nanostar in Fig. 3 happen to coincide with the 3PPE-to-4PPE cross-over around 830 nm (Fig. 1e), the observed behaviors are a general feature of multi-resonance geometries and are also demonstrated in the next section with a nanostar studied entirely in the 3PPE regime.

**Frequency- and polarization-controlled photoemission.** The nanostar in Fig. 4a displays simple multi-resonance behavior, with a higher-energy (blue) dipolar resonance mode aligned with the shorter tip and a lower-energy (red) dipolar resonance mode aligned with the longer tip. These two tips are approximately orthogonal and can be individually addressed, as established in the polarization dependence at different excitation frequencies (Fig. 4b). Instead of maintaining the circular polarization, tip selectivity can be achieved with frequency tuning alone, as

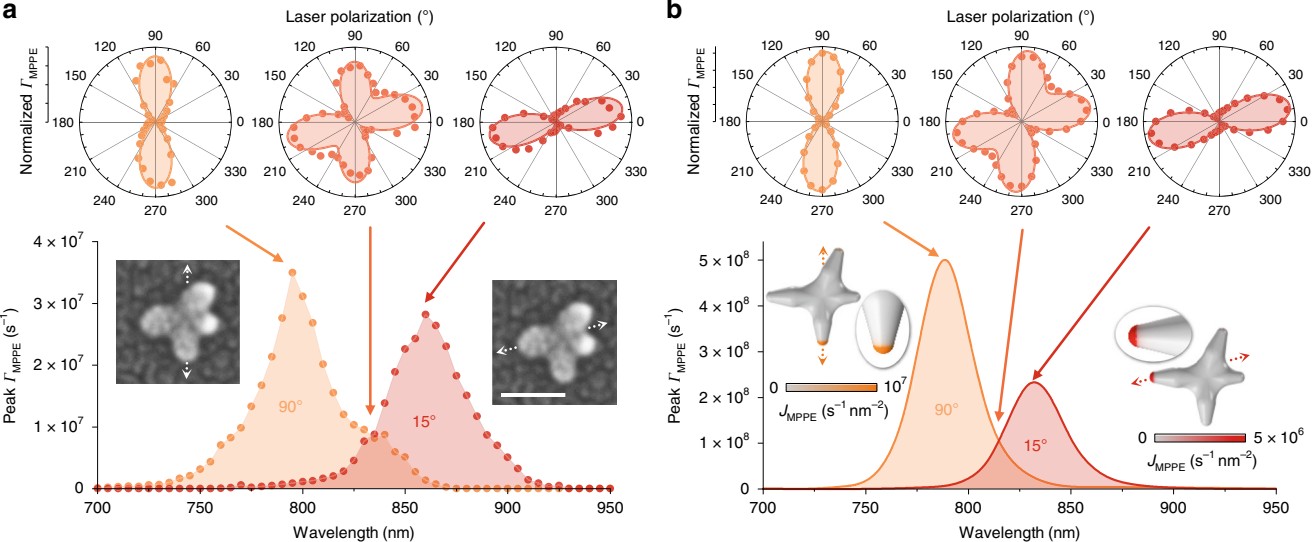

**Fig. 3 Multi-tip excitation and selectivity. a** Experimental slices through the optical frequency and linear polarization parameter space, with photoemission peaks at (795 nm, 90°) and (860 nm, 15°). At either spectral peak, the laser polarization dependence shows minimal signature of the other resonance, while between resonances (835 nm) the four-lobed behavior shows equal contributions from both resonance modes. The peak polarization axes are indicated in the correlated SEM insets for each resonance, as determined via nonlinear cosine fits shown in the polarization plots (50 nm scale bar). **b** Theoretical surface multiphoton photoemission rates for the correlated nanostar geometry. Aside from a small blue shift, peaks at (785 nm, 90°) and (830 nm, 15°) are in good agreement with the experimental results. All four tips are modeled with the same radius (3.4 nm), but photocurrent density maps demonstrate single-tip excitations at the resonance conditions due to the specific plasmonic mode structure. Measurements and calculations are performed at $1 \times 10^8$ W cm$^{-2}$ peak input intensity.

discussed in the previous section and further demonstrated here via MPPE simulations (Fig. 4c). Velocity distributions measured (Fig. 4d) and calculated (Fig. 4e) at each frequency are directionally aligned with the corresponding resonant tip axis and photoemission directionality is rotated by a full 90° upon frequency-controlled switching between tips. The average electron kinetic energy decreases with excitation frequency by conservation of energy. When both tip modes are excited at intermediate frequencies (e.g., Fig. 4c, 775 nm), the resulting velocity distribution is simply a linear combination of the individual tip angular distributions. Thus, this linear combination allows for a continuous steering of the average emission angle, although the total angular distribution is broadened by arising from two separate point-like sources.

Polarization-controlled directional emission is presented in Fig. 5 for the same nanostar as in Fig. 4, but exclusively at the intermediate frequency setting at which both tips are equally resonantly enhanced. Simulations demonstrate switching of photoemissive regions between the two tip hot spots as the linear polarization is rotated out of alignment with one mode and into alignment with the other (Fig. 5a). The photoemission directionality is rotated by 90° (Fig. 5b, c) in the same manner observed via frequency control, due to the same underlying process of selective hot-spot excitation. Intermediate polarizations again result in a linear combination of the two tip angular distributions, with the relative weights determined by the polarization dependence (Fig. 5a). Although full polarization contrast is demonstrated by complete alignment along either tip mode, the polarization plot indicates that much less angular tuning is necessary to strongly favor one tip over the other due to the $\cos^6(\theta - \theta_{tip})$ polarization sensitivity for each plasmon mode. Strong tip selectivity can be achieved with a 90:10 emission ratio by only ±10° tuning away from the intermediate polarization, at which the emission ratio is 50:50. Therefore, as a benefit of the MPPE nonlinearity, photoemission directionality can be rotated by 90° with only 20° polarization rotation. This

fine degree of tip discrimination also indicates possibilities for utilizing higher tip densities for a more continuous control of angular photoemission.

## Discussion
Versatile photoemission switching/steering has been demonstrated by independently tuning one of two optical degrees of freedom (frequency or linear polarization), leaving the other available for modifying the control characteristics. For example, polarization can be utilized for tip selection and corresponding manipulation of angular currents, while frequency can be simultaneously utilized to control the electron kinetic energy distributions and relative tip photoemission enhancement. Such possibilities illustrate how techniques developed for coherent control over nanoplasmonic hot spots using femtosecond optical amplitude, frequency (phase), and polarization shaping[18,45] can be directly applied to photocurrent degrees of freedom. The photoemission switching timescale is fundamentally limited by the plasmon dephasing ($T_2$) time and by the optical cycle of the excitation laser field, which defines the fastest timescale on which polarization and frequency can be manipulated. Due to the $E^{2n}$ process nonlinearity, the $n$PPE current decays $2n$-times faster than the plasmonic field, which typically dephases in 10 fs or less[33,46]. Thus, the 3PPE and 4PPE decay times are comparable to the 1–3 fs optical cycles for visible and near-infrared frequencies. This suggests that spatiotemporal control over hot-spot excitation and directional current generation may be achieved on timescales approaching the attosecond range, even in the weak-field MPPE intensity regime.

Gold nanostars behave as prototypical nanoplasmonic cathodes, with the multi-tip geometries shown to provide direct maps between laser parameters and excitation of different hot spots. Individual tip hot spots have been extensively characterized for sample nanostars via polarization and spectral studies, correlated SEM imaging, and finite element simulations. Angle-resolved

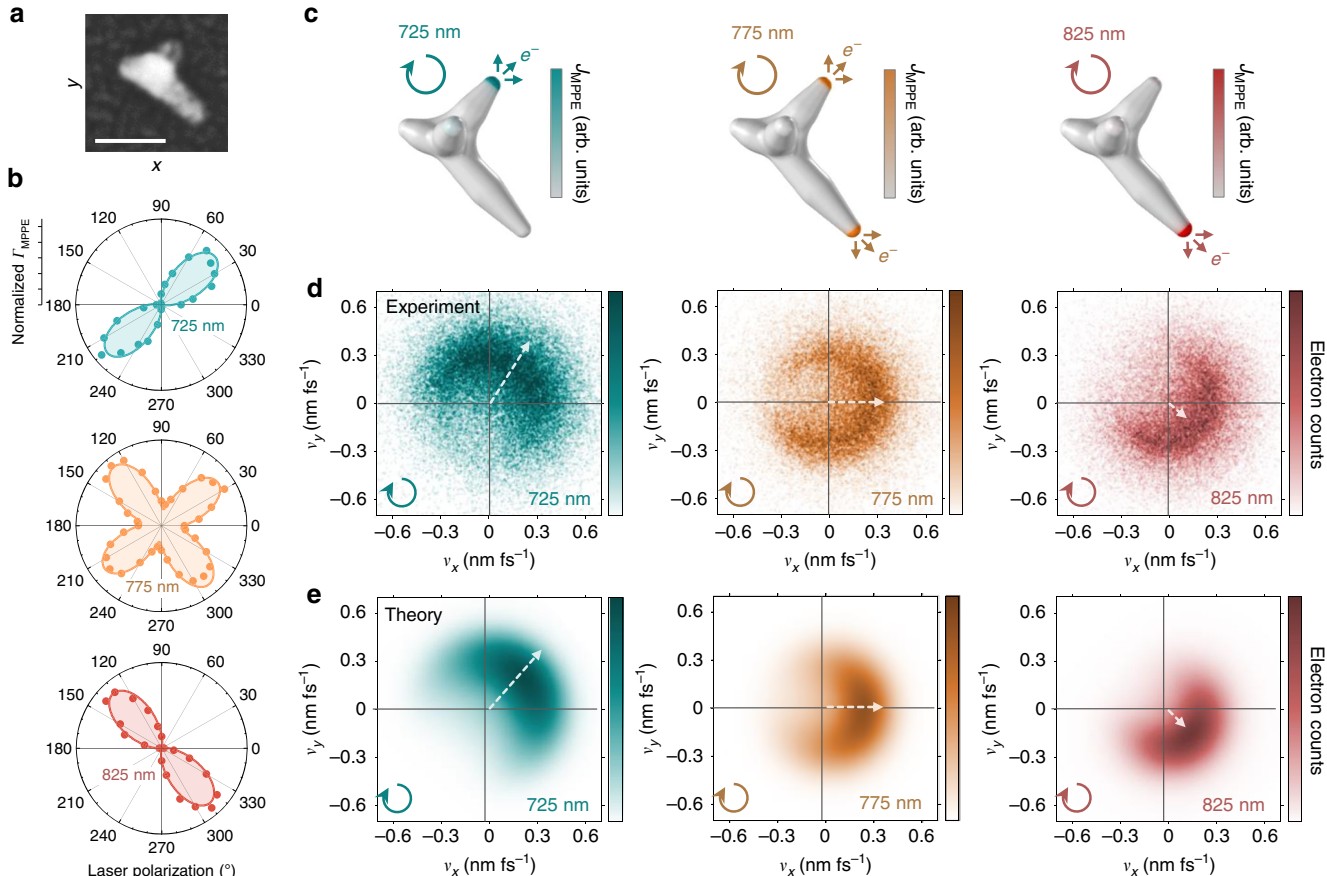

**Fig. 4 Frequency-controlled directional photoemission. a** Correlated nanostar scanning electron micrograph with the shorter and longer resonant tips at 50° and 315°, respectively (50 nm scale bar). **b** Experimental polarization dependence for frequency at the short-tip resonance (725 nm, blue), between resonances (775 nm, orange), and at the long-tip resonance (825 nm, red) with nonlinear cosine fits for the 3-photon process. **c** Calculated multiphoton surface current distribution at different frequencies, showing the transition from one tip hot spot to the other with circular polarization. **d** Experimental and **e** theoretical $v_z$-projected velocity maps on and between the two resonances using circular polarization. Vectors indicate the peak directions determined by Gaussian fits around the peaks of the angular distributions. Peak photoemission angles range from 55° (50°) at 725 nm to 325° (315°) at 825 nm for the experimental (theoretical) velocity distributions, i.e. a 90° photoemission rotation. The vector magnitude represents the speed ($v_F$) of photoelectrons excited from the Fermi level, which decreases with decreasing photon energy by energy conservation, $\frac{1}{2}m_e v_F^2 = n\hbar\omega - \phi$, with tip work function $\phi = 4.5$ eV and $n = 3$ in this excitation energy range ($\hbar\omega = 1.5$–1.7 eV). The Fermi velocity does not coincide with the apparent edge of the 825 nm (red) distribution due to the velocity dependence of the photoemission amplitude, which leads to deviations from a simple Fermi–Dirac distribution (particularly near-zero velocity) and shifts the effective edge outward relative to $v_F$ for near-threshold photon energies.

photoelectron velocity measurements demonstrate corresponding frequency and polarization control over photoemission current direction, with all experiments corroborated by 3D surface-mediated photoemission calculations that can be carried out for arbitrary nanoplasmonic geometries. Although volume-mediated excitation processes must also be considered in general, the observed directionality and agreement between experiment and theory strongly underscores the dominant role of surface-mediated MPPE at the sharp nanostar tips. The results presented here highlight opportunities for implementing designer plasmonic nanoparticles and nanostructures as all-optical photocurrent control elements in a variety of applications, including femtosecond electron imaging and diffraction, polarization-sensitive photodetection, site-selective photocatalysis, and terahertz nanoelectronics.

## Methods

**Nanostar synthesis and sorting**. Nanostars are synthesized via a seedless growth method, in which HEPES buffer functions both as a nucleation and a shape-directing agent. A 1 M stock HEPES solution is made by dissolving the buffer salt in Millipore water (18.2 MΩcm). The pH of the HEPES solution is measured and adjusted to 7.38 using concentrated NaOH solution. Nanostars are synthesized by adding 0.2 mM (final concentration) gold (III) chloride trihydrate (HAuCl₄) to 110 mM HEPES buffer and vortexing for 1 min after HAuCl₄ addition. The growth solution is left in the dark for 24 h to allow for growth and stabilization. To improve size homogeneity and to achieve the desired resonances within the tuning range of Ti:sapphire laser, density gradient centrifugation is carried out to sort the nanostars. A continuous linear sucrose gradient is created by using a custom mixing program of the gradient maker (BioComp Instruments) to mix 50% and 60% weight-to-volume sucrose solutions at an angle. At this point, 500 μL of 8–10 nM concentrated nanostar solutions are layered on the top of the prepared gradient and centrifuged at 4400 × *g* for 4 h. The samples are fractionated at intervals of 4 mm from the meniscus (BioComp Instruments) and centrifuged at 15,000 r.p.m. for 25 min to remove excess sucrose. Each fraction is then dialyzed in Thermo Fisher 20K Slide-A-Lyzer Dialysis Cassettes for 24 h to remove remaining sucrose from the solution. TEM images are taken of each fraction to choose the population with desired size and aspect ratio of the branches. Fraction 7 out of 23 is selected for the present studies.

**Sample preparation**. Nanostar samples are prepared on glass coverslips sputter coated with a 10 nm ITO film, which provides ohmic conductivity for charge neutralization along with visible transparency for back-side illumination and excitation of the nanostars. A 50 nm alphanumeric gold grid is patterned onto the ITO via negative photomask lithography using an uncoated copper TEM grid

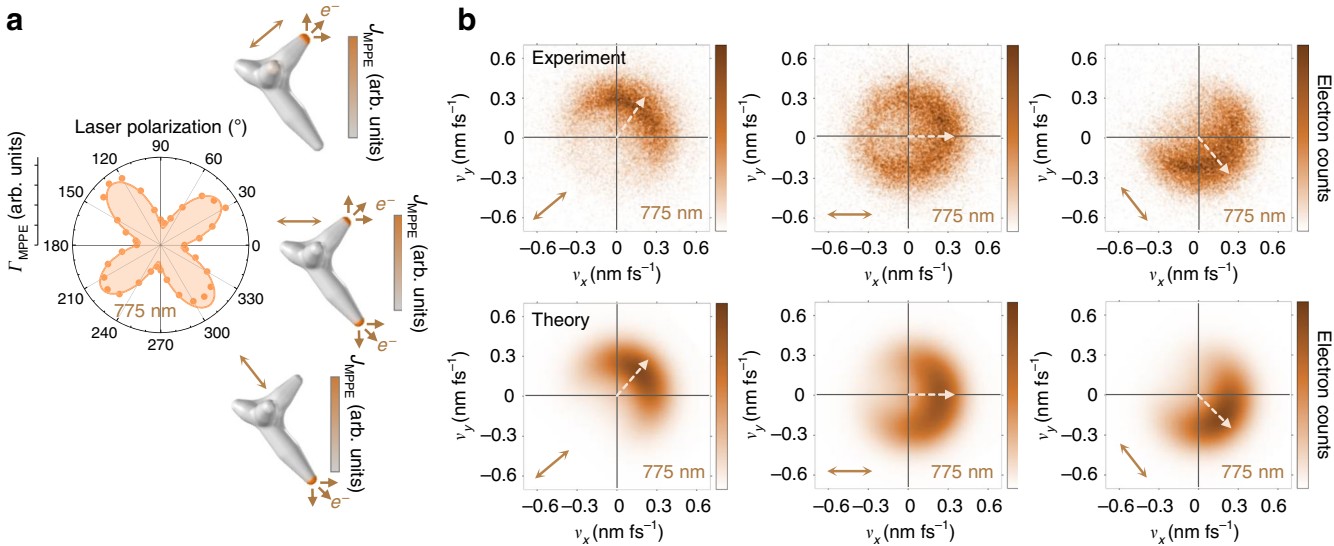

**Fig. 5 Polarization-controlled directional photoemission. a** Experimental nanostar polarization dependence at 775 nm with nearly equal contributions from both nanostar tip resonances. Tip hot spots are selected by rotating linear polarization while keeping frequency constant, as demonstrated in the calculations on and between the resonances. **b** Experimental and **c** theoretical $v_z$-projected velocity maps for the different polarizations at constant frequency. Vectors indicate the peak directions determined by Gaussian fits around the peaks of the angular distributions. Peak photoemission angles range from 55° (50°) at 50° linear polarization aligned along the short tip to 310° (315°) at 315° linear polarization aligned along the long tip, for the experimental (theoretical) velocity distributions. The vector magnitude represents the velocity of photoelectrons excited from the Fermi level ($v_F$), which remains constant for all distributions due to the fixed excitation frequency.

(LF-400) as the mask, enabling particle locating in correlated SEM-SPIM (scanning photoelectron imaging microscopy) studies. The ITO substrates are ultraviolet ozone cleaned to remove hydrocarbons and to permit surface wetting, after which a 50 μL aliquot of aqueous nanostar solution is spin coated for 5 min on the substrate at 1500 r.p.m. The dilution is optimized for a final particle coverage of ~0.05 nanostars μm$^{-2}$, such that ~20 nanostars can be studied in a $20 \times 20$ μm$^2$ SPIM scan, with negligible probability of two nanostars overlapping within the 500 nm diffraction-limited laser spot.

**Scanning photoelectron imaging microscopy**. The SPIM technique combines scanning photoemission microscopy/spectroscopy for single-particle characterization with a VMI electrostatic lens for transverse ($v_x$, $v_y$) photoelectron velocity mapping. Femtosecond pulses are generated via a 75 MHz Ti:sapphire oscillator (KMLabs Swift, 700–1000 nm), with second harmonic generation (350–500 nm) and an optical parametric oscillator (KMLabs, 510–780 nm signal output tuning range) providing broad tunability throughout the visible and near-IR. For optimal frequency tunability without spatial walk-off, no external prism dispersion compensation is utilized for the majority of the present studies. Group velocity dispersion in the system thus results in 100–200 fs pulse durations at the sample across the laser tuning range. As a check, several measurements were also performed with dispersion-compensated (~50 fs) pulses and found to yield indistinguishable photoelectron velocity distributions. A high-vacuum-compatible ($<5 \times 10^{-7}$ Torr) reflective microscope objective (NA = 0.65) focuses the pulsed laser beams to a diffraction-limited spot (~500 nm) on the sample in a normal-incidence, back-illuminated configuration (Fig. 1b).

The ITO-coated sample coverslip is situated on a scanning copper stage, both of which are held at −4500 V and together serve as the repeller electrode in the VMI stack (see Fig. 1b). The VMI lens provides a linear velocity-to-position mapping onto a spatially resolved chevron microchannel plate electron multiplier, in which a single electron is multiplied up to $10^7$ electrons. The amplified electron signal is then accelerated onto a P47 phosphor screen, the fluorescence from which is imaged on a 1.3 megapixel CCD camera running at 20 frames per second. Single event (x, y) coordinates are determined via centroiding and converted to ($v_x$, $v_y$) velocity with a calibration factor of 4150 m s$^{-1}$ px$^{-1}$ measured previously[27]. Three quartered piezoelectric posts with capacitive sensor feedback provide fine scanning over a $30 \times 30$ μm$^2$ area, with $xyz$ piezo motors providing extended positioning over larger millimeter distances. Single plasmonic nanoparticles can thus be identified and studied for hours or days with minimal drift correction. The nanostars are cleaned prior to all studies via brief (~1 s) exposure to ~1 GW cm$^{-2}$ of second harmonic light at 400 nm, which removes adlayers (i.e., water) that develop during exposure to ambient air[25].

**Transmission and scanning electron microscopy**. Nanostars are examined via TEM (FEI Tecnai T12 SpiritBT, 100 kV, LaB$_6$) after drop casting 15 μL of the aqueous nanostar solution onto a carbon-coated TEM grid for 10 min, removing

the excess solution, and drying in air. Correlated SEM studies (FEI Nova NanoSEM 630, 10 kV, through lens detector, field immersion mode) are performed on the glass/ITO sample substrates following SPIM studies using the Au grid reference to locate the same nanostars in both systems. The sample is coated with a 3 nm Pt film prior to imaging to enhance conductivity, thereby improving contrast and reducing degradation due to charging and carbon buildup during electron beam exposure. Refer to Supplementary Fig. 1 for TEM and SEM statistical characterization of the nanostar dimensions.

**Surface MPPE modeling**. We implement the theory developed by Yalunin et al.[40] to model surface-mediated MPPE velocity distributions for coherent excitation at a metal–vacuum interface, modified into a two-step process to account for the high-gradient evanescent surface plasmonic fields. In the first step, free-electron initial states are excited into field-dressed final states (Volkov states) with the ponderomotive quiver energy appearing in the energy conservation equation,

$$\frac{\hbar^2 k^2}{2m_e} + n\hbar\omega = \frac{p^2}{2m_e} + U_p + E_F + \phi, \qquad (3)$$

in which $\hbar k$ is the initial state momentum, $p$ is the final state drift momentum, $U_p$ is the ponderomotive energy, $E_F$ is the Fermi energy, and $\phi$ is the work function. In the second step, as the electron leaves the evanescent surface plasmon field, the ponderomotive energy is fully converted into kinetic energy corresponding to the surface-normal momentum. The differential multiphoton photocurrent is given by

$$\frac{dJ_{MPPE}}{d^3k} = \frac{2\hbar}{(2\pi)^3 m_e} \sum_{n \geq n_{min}} \frac{k_z}{e^{(\hbar^2 k^2/2m_e - E_F)/kT} + 1} P_n(k_z), \qquad (4)$$

which is the collision rate of Fermi sea electrons on the surface potential barrier, scaled by the dimensionless excitation and emission probability, $P_n(k_z)$, and summed over all allowed multiphoton process orders as determined by energy conservation.

The photoemission rate can be written in terms of external momentum via coordinate transformation

$$\hbar k_x = p_x, \qquad (5a)$$

$$\hbar k_y = p_y, \qquad (5b)$$

$$\frac{\hbar^2 k_z^2}{2m_e} + n\hbar\omega = \frac{p_z^2}{2m_e} + U_p + E_F + \phi. \qquad (5c)$$

The Jacobian determinant is $\left| d^3k/d^3p \right| = p_z/(\hbar^4 k_z)$ and the photoemission probability is thus given in external momentum coordinates by

$$\frac{dJ_{MPPE}}{d^3p} = \frac{2}{h^3 m_e} \sum_n \frac{p_z}{e^{(p^2/2m_e + U_p + \phi - n\hbar\omega)/kT} + 1} P_n(k_z(p_z)), \qquad (6)$$

in which all process orders may contribute to a final momentum state at finite

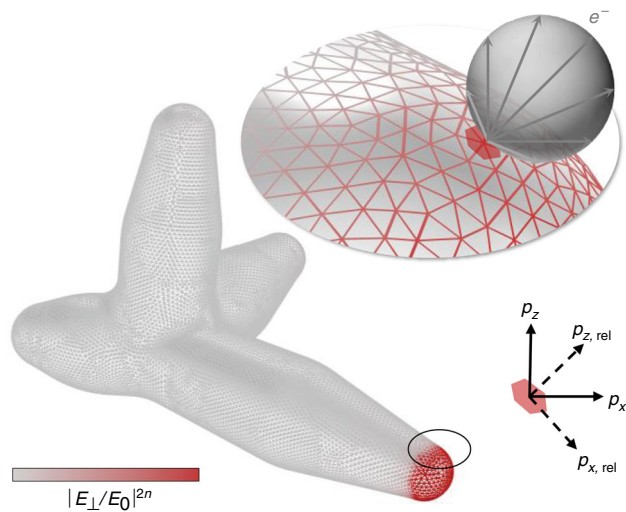

$$|E_\perp / E_0|^{2n}$$

**Fig. 6 Surface photoemission simulation geometry.** Example nanostar surface mesh and nonlinear field enhancement distribution utilized in theoretical 3D photoemission calculations. Multiphoton photoemission distributions are calculated with respect to the surface normal for each nanostar surface area element (e.g., shaded patch) corresponding to each triangular mesh vertex at which the surface field enhancements are calculated.

temperature due to the exponential Fermi–Dirac tail, although most processes are negligible at room temperature except the dominant multiphoton order dictated by energy conservation. The photoemission probability derived by Yalunin et al.[40] via Green's function solution to the time-dependent Schrödinger equation is given here in terms of external coordinates as

$$P_n(k_z(p_z)) = \frac{\hbar^2}{\sqrt{2m_e}(E_F + \phi)} \frac{\sqrt{\frac{p_z^2}{2m_e} + U_P + \phi - n\hbar\omega}}{p_z} \left| I_n(p_z) + R_n^{(1)} R_n^{(2)} I_n(-p_z) \right|^2, \quad (7a)$$

$$I_n(p_z) = \frac{\sqrt{2m_e}}{h} \int_0^{2\pi} \left( i\sqrt{n\hbar\omega - U_P - \frac{p_z^2}{2m_e}} + \frac{p_z}{\sqrt{2m_e}} - \sqrt{2U_P} \right) e^{iS(q)} \, dq, \quad (7b)$$

$$S(q) = nq + 2\frac{p_z}{\hbar\omega}\sqrt{\frac{U_P}{m_e}}\cos(q) - \frac{U_P}{2\hbar\omega}\sin(2q), \quad (7c)$$

$$R_n^{(1)} = -\frac{\sqrt{\frac{p_z^2}{2m_e} + U_P + E_F + \phi} - \frac{p_z}{\sqrt{2m_e}}}{\sqrt{\frac{p_z^2}{2m_e} + U_P + E_F + \phi} + \frac{p_z}{\sqrt{2m_e}}}, \quad (7d)$$

$$R_n^{(2)} = J_0\left(-4\frac{p_z}{\hbar\omega}\sqrt{\frac{U_P}{m_e}}\right). \quad (7e)$$

The integral terms $I_n(\pm p_z)$ represent the outward- and backward-moving excited waves outside the medium. The reflection coefficients on the backward-moving wave account for rescattering on the surface potential barrier, where $R_n^{(1)}$ is the reflection coefficient for a step-down potential (height $E_F + \phi$) and $R_n^{(2)}$ accounts for the effect of the oscillating triangular barrier in the applied optical field. Interference between the direct and rescattered waves can have significant effects on the final emission amplitude, as discussed in detail by Yalunin et al.[40]. Finally, we include the ponderomotive energy transfer in the present case of an evanescent plasmonic field via the coordinate transformation $p_z^2/2m_e \rightarrow p_z^2/2m_e - U_P$.

To calculate the full 3D photoelectron velocity distribution for a given nanostar, the photoemission contributions from each surface area element are calculated using the surface field enhancements determined via finite element simulation. The photoemission distribution is calculated with respect to each surface normal ($\hat{p}_{z,\text{rel}}$) and rotated into the global frame ($\hat{p}_z$) via Cartesian rotation matrices. Only a single rotation about axis $\hat{p}_z \times \hat{p}_{z,\text{rel}}$ is required for azimuthally-isotropic distributions. A sample uniform surface mesh and nonlinear surface-normal field enhancement distribution is shown in Fig. 6, with additional details of the finite element domain presented in Supplementary Fig. 2. Empirical values are utilized for all model

inputs, with the exception of the Fermi–Dirac electron temperature, which is calculated at 2500 K via a two-temperature model (Supplementary Fig. 6 and Supplementary Note 4). It should finally be noted that geometries with concave surface regions (including nanostars) may allow for the intersection of emitted electrons with other surfaces of the emitter geometry. These effects are not presently accounted for, although they should be negligible for the tip-like emission described in the present work.

## Data availability
All data required to reproduce the findings in this study are available from the corresponding authors upon reasonable request.

## Code availability
The MATLAB script for surface-mediated multiphoton photoemission calculations is available from the corresponding authors upon reasonable request.

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

## Acknowledgements

We would like to acknowledge Dr. K. Culver's contributions to the nanostar synthesis process along with her role in stimulating these investigations. J.P., F.M., and D.J.N would also like to thank Prof. M. Müller and Prof. M. Raschke for illuminating conversations. Photoemission studies (D.J.N.) were supported by the Air Force Office of Scientific Research (FA9550-15-1-0090) with additional funds for laser development and apparatus construction provided by the National Science Foundation (CHE 1665271, PHY 1734006). Nanostar synthesis (T.W.O.) was supported by the National Science Foundation (CHE 1808502).

## Author contributions

J.P., F.M., and D.J.N designed the experiments. P.C. and T.W.O. developed the synthetic methods, synthesized the gold nanostars, and characterized the samples. J.P. and F.M. performed the experiments, analysis, and calculations. All authors contributed to the writing of the manuscript and provided approval of the final version.

## Competing interests

The authors declare no competing interests.
