## [Peer Review File · Nature Communications]

Reviewers' Comments:

Reviewer #1:

Remarks to the Author:

The authors study photoemission from gold nanostars. They show, by means of angle-resolved 2D velocity mapping that the angular photocurrent distribution can be controlled by varying the polarization and frequency of the driving laser. The experiments are carefully performed and the results are clear and interesting. Essentially, the authors show that by varying the laser parameters, they can control which tip of the nanostars emits the photoelectrons. The interpretation of experimental results is strongly supported numerical simulations of the photoemission process based on a model introduced earlier by Sergey Yalunin and coworkers.

Even though the results may not appear extraordinarily surprising, the results are interesting and may merit publication in Nature Communications. Before recommending publication, I would like to ask the authors to carefully address the following questions:

1. What information can be deduced from these measurements about the local field enhancement at the apex of the nanostars?
2. I found it surprising to see that the nonlinear order n of the photoemission (Fig. 1e) does not change with peak laser intensity. Are there signatures of strong-field photoemission in these measurements? If not, why not?
3. The nanostars have a fairly small radius of curvature at their apex. How sensitive are these objects to laser-induced melting or structural damage. What is the lifetime of the nonlinear nanostar photoemitters?
4. The experiments are performed with a 75 MHz laser system and electron counts of more than 10^7 s⁻¹ are reported in Fig. 1e. Hence, more than 10 electrons are emitted per pulse. How does the Coulomb interaction between these electrons affect the angular emission pattern?
5. The excellent agreement between experimentally observed photoemission maps is surprising and merits further discussion. I believe that the manuscript would benefit from discussing the physics underlying these simulations in more detail in the main manuscript. What exactly is the geometry of the nanostars that has been taken into account in the simulations? How does the field distribution look like? Has the substrate been taken into account in the simulations? Which energy/momentum distribution of the electrons inside the tip is considered? Does the model consider the substantial heating of the electron gas during the optical excitation? If not: is the approximation of a cold electron gas justified? What happens to the electrons that recollide with the tip? How does the model treat the electrons that are driven into the metal? What is the quiver length of the nanostar near-field? A small sketch that describes the simulation geometry, the field distribution, the momentum distribution of the electrons inside the tip and selected trajectories of the photoemitted electrons would greatly help to understand the basic assumption of the simulations. Giving just the main formulas in the Appendix is not sufficient.
6. Bainbridge and Bryan (NJP, 2014) have reported on velocity map imaging of photoelectrons emitted from nanotips and Park et al. (PRL, 2012) have reported on strong-field effects in the angular distribution of photoemitted electrons from metal tips. Those papers might be cited.

Reviewer #2:

Remarks to the Author:

Referee report on article NCOMMS-19-32909

Plasmonic Nanostar Photocathodes for Optically-Controlled Directional Currents

by Pettine, Nesbitt et al.

The paper NCOMMS-19-32909 addresses a topic of high current interest, namely the optical control over directional currents with the potential of fs time resolution. The authors bring in their excellent expertise in the field of nanoscale plasmonic resonators and emitters. The paper is well written and scientifically sound. The data are of high interest to a wide readership in the field of ultrafast spectroscopy, diffraction and microscopy, terahertz technology and even accelerator

physics.

Two major and some minor points of criticism are listed below.

I recommend publication in Nature Communications after the authors have met my major points and addressed or commented on the minor points.

Major points:

1. The authors discuss the intensity dependence for a specific star as follows:

Log-log intensity dependence plots and power-law fits are shown for the three sample nanostars, with the Star 2 resonance ($n = 3.6 \pm 0.1$) representing an intermediate process with weighted contributions from both 3PPE and 4PPE.

If indeed weighted contributions from 3PPE and 4PPE occur for Star 2, it must be considered that the two contributions have a different power dependence. Mathematically, the logarithm of a sum of two power terms (Eq. 1) does not result in a linear dependence in the double-logarithmic plot. The authors should address this mathematical argument, re-examine these results and explain the linear dependence shown in Fig. 1e, star 2 more carefully. Maybe one of the contributions is negligible so that the deviation from linear is very small? But that would not fit to $n=3.6$

To me, something seems wrong here.

2. Is the dI/dE dependence shown in Figure 2h really compatible with a 3PPE process? I would have expected a differently-shaped function. For the derivative one would expect a negative value on the falling wing, because the intensity decreases with increasing kinetic energy. Maybe it does not show the derivative but simply the spectrum? Moreover, I would find it very helpful to assist the reader with a few sentences about the role of the ponderomotive forces and how much they actually contribute to the kinetic energy in Figure 2h. This must be strongly depend on the incident light intensity, a fact that should be stated in the text.

Minor points:

3. The wet-chemical growth and subsequent dialysis might leave residues of solvent or sucrose on the surface of the nanostars. An ultrathin insulating (mono-)molecular layer between one of the tips of the nanostar and the conductive substrate might give rise to gap resonances (as discussed in Phys. Rev. Lett. 108 (2012) 237602). This would explain the strength of emission and the high finesse of the nanostar resonators. It might also explain why the tips pointing away from the surface obviously do not emit (at least not significantly). Since some of these are strongly tilted, the photon polarization argument alone cannot explain this. Due to irregularities in the shape, such gap resonances could possibly also occur even without an insulating spacer layer.

Although to some extent speculative, I feel that the authors should at least mention the possible presence of gap resonances.

4. In the context of femtosecond emitters the dephasing time (T_2) certainly plays a role. Could the authors please comment on how T_2 enters into the controlled emission process. The "ringing plasmon" might pose a limitation on the achievable time structure. Or maybe I overlook something here?

5. The authors might want to rephrase a sentence in the abstract: In the nano-world, R 3.4 nm is sharp but not "point like".

6. My last point concerns the way, how a closely-related previous publication is mentioned.

The abstract states: "However, angular photocurrent distributions in nanoplasmonic systems remain poorly understood..."

And on p.4: "Such capabilities have only appeared recently (24,26-28)..."

I feel that the results of ref. 27 (Nano Lett. 17, 6606-6612 (2017)) should be mentioned in some more detail as in the short remark on p.4. That paper deals with the closely-related issue of plasmonic emission from Au nanorods on an ITO substrate, with backside illumination with fs-pulses from a Ti:sapphire laser using a momentum microscope that also resolves individual nanorods in real-space imaging mode. This was actually the first study of the momentum- and energy-distribution of plasmonic electron emission from resonantly-excited, individual, selected Au nanorods. The directional emission from the tips (with the rods lying flat on the surface, identical to the emitting branches of the nanostars) has been resolved not only on a true momentum scale (calibrated in inverse Angstroms) but also on the energy scale, giving valuable additional information on the emission process.

The present work on nanostars is excellent and the paper is very convincing. However, the fact

that a gold nanotip pointing parallel to the surface is a highly efficient plasmonic emitter has been shown in ref. 27.

I list this under the minor points and leave it to the authors how far they implement a fair judgement of the ref. 27 work.

Reviewer #1 (Remarks to the Author):

The authors study photoemission from gold nanostars. They show, by means of angle-resolved 2D velocity mapping that the angular photocurrent distribution can be controlled by varying the polarization and frequency of the driving laser. The experiments are carefully performed and the results are clear and interesting. Essentially, the authors show that by varying the laser parameters, they can control which tip of the nanostars emits the photoelectrons. The interpretation of experimental results is strongly supported numerical simulations of the photoemission process based on a model introduced earlier by Sergey Yalunin and coworkers.

Even though the results may not appear extraordinarily surprising, the results are interesting and may merit publication in Nature Communications. Before recommending publication, I would like to ask the authors to carefully address the following questions:

1. What information can be deduced from these measurements about the local field enhancement at the apex of the nanostars?

While we do not directly measure the near field, the measured photoelectron angular distributions are highly sensitive to the plasmonic field and can provide a fine level of detail on the tip hot spots when combined with finite element modelling and multiphoton photoemission theory. The agreement between experimental and theoretical angular and kinetic energy distributions for a single representative nanostar tip (e.g. Fig. 2) and for multi-tip relative field enhancements (Figs. 3-5), along with the level of quantitative agreement demonstrated in Fig. 3 (corresponding to fields within 50% of the experimental value), indicate a reasonably faithful reproduction of both the plasmonic near-field distributions and magnitudes.

The field magnitude is now addressed in further detail on p. 16 of the main text. To further clarify the hot spot field distribution for the representative nanostar studied in Fig. 2 – which we scrutinize most closely for single-tip emitter properties – we have added top- and side-view domain slices of the simulated field enhancements to Supplementary Fig. 2.

2. I found it surprising to see that the nonlinear order n of the photoemission (Fig. 1e) does not change with peak laser intensity. Are there signatures of strong-field photoemission in these measurements? If not, why not?

We have chosen conditions to ensure that all of photoemission occurs within the multiphoton regime for these studies, as described in the following locations:

- First, as motivation for working in the MPPE regime we explain at the end of the introduction that “weak-field multiphoton photoemission (MPPE) processes are emphasized here due to the minimal nanostar heating, sub-single-electron femtosecond pulses ($\sim 10^{-5} e^-$ /pulse) that preclude space-charge effects, and < 1 eV photoelectron kinetic energy spreads for high temporal coherence.”
- As noted by the reviewer, the MPPE emission mechanism is demonstrated by the strict $n = 3$ and $n = 4$ power-law dependences measured in intensity-dependence plots in Fig. 1e, as well as in the clear transition between these process orders with laser frequency, none of which is consistent with strong-field or thermionic emission. This does not preclude small contributions from these

processes, but if we were near either of these regimes we would expect to see indication of this in the intensity dependence for at least some of the nanostars.

- On p. 8, we calculate the Keldysh parameter using conservative bounds for the plasmonic hot spot field enhancements ($|E/E_0| < 100$) and peak input laser intensities ($I_0 < 3 \times 10^8 \text{ W/cm}^2$), yielding Keldysh parameters > 3 in the MPPE regime. With typical simulated field enhancements ($|E/E_0| \approx 50$) and input intensity ($I_0 \approx 2 \times 10^8 \text{ W/cm}^2$), $\gamma \approx 8.5$ is well within the MPPE regime.
- Measured photocurrents are orders of magnitude lower than in typical nanoparticle optical field emission studies (e.g. Hobbs & Berggren, *ACS Nano* (2014)), in which currents approach or exceed one electron per femtosecond pulse. By contrast, we only emit $\sim 10^{-5} e^-/\text{pulse}$ – also see response to Remark 4 for further discussion.

3. The nanostars have a fairly small radius of curvature at their apex. How sensitive are these objects to laser-induced melting or structural damage. What is the lifetime of the nonlinear nanostar photoemitters?

We observe little/no evidence of tip melting in our MPPE experiments, in which the nanostar lifetimes are much longer than our measurements (collected over hours on single nanostars in some cases). We have now emphasized this stability on p. 13 of the manuscript, noting a modest calculated temperature increase of $\Delta T_{\text{lattice}} \approx 200 \text{ K}$ for the utilized intensities. To further address the temporal stability of these tips, we have also added a new figure to the supplementary information (now Supplementary Fig. 5) showing the photoemission over time for the three representative single-tip emitters (Stars 1-3) and three other stars.

As additionally noted in Supplementary Fig. 1, no observable tip morphology changes occur following typical laser exposure conditions within our SEM resolution for the uncoated micrographs presented in Supplementary Fig. 1a (middle row), compared with the TEM micrographs of unexposed nanostars (top row).

4. The experiments are performed with a 75 MHz laser system and electron counts of more than 10^7 s^{-1} are reported in Fig. 1e. Hence, more than 10 electrons are emitted per pulse. How does the Coulomb interaction between these electrons affect the angular emission pattern?

This is a crucial aspect of our experiment and we appreciate the opportunity for clarification. The photoemission currents reported are *peak* photocurrents occurring at peak laser pulse intensity. We therefore multiply the peak photocurrents ($\sim 10^8$ electrons/s) by the pulse duration ($\sim 10^{-13}$ s/pulse) to arrive at $\sim 10^{-5}$ electrons/pulse, i.e. negligible probability of even two electrons ever being emitted in a single pulse and therefore no Coulomb repulsion/distortion.

We have now added a more detailed discussion to the text on p.8 to clarify this point, and also note this at the end of the Introduction by explicitly mentioning “sub-single-electron femtosecond pulses ($\sim 10^{-5} e^-/\text{pulse}$) that preclude space-charge effects”.

5. The excellent agreement between experimentally observed photoemission maps is surprising and merits further discussion. I believe that the manuscript would benefit from discussing the physics underlying these simulations in more detail in the main manuscript. What exactly is the geometry of the nanostars that has been taken into account in the simulations? How does the field distribution look like? Has the substrate been taken into account in the simulations? Which energy/momentum distribution of the electrons inside the tip is considered? Does the model consider the substantial heating of the electron gas

during the optical excitation? If not: is the approximation of a cold electron gas justified? What happens to the electrons that recollide with the tip? How does the model treat the electrons that are driven into the metal? What is the quiver length of the nanostar near-field? A small sketch that describes the simulation geometry, the field distribution, the momentum distribution of the electrons inside the tip and selected trajectories of the photoemitted electrons would greatly help to understand the basic assumption of the simulations. Giving just the main formulas in the Appendix is not sufficient.

We thank the reviewer for highlighting this opportunity for clarification. We have addressed these points with the following additions to the text:

- Further details of the modelled nanostar geometries are now provided in the main text (p. 9) and in Supplementary Note 1.
- In the caption for Fig. 2b and on p. 9 of the main text, we now explicitly note that the ITO-coated substrate and HEPES ligands are included in the field enhancement simulations.
- We now include a much more extensive exposition of the theory and the approximations therein in the main text (p. 11-12), including the two-temperature model electron heating calculations, the MPPE excitation mechanism at the surface, consideration of direct/rescattered quantum pathways, and electron decay length + quiver amplitude + plasmonic field decay effects on ponderomotive acceleration.
- We have added a Fig. 6 to the Methods section to better illustrate the effects of geometry, meshing, and nonlinear field enhancement in the theoretical calculations for the interested reader, while not distracting from the main conversation.
- Additional domain sections have been added to Supplementary Fig. 2 for further detail on the calculated near-field enhancement distribution.
- We now note on p. 26-27: “It should finally be noted that geometries with concave surface regions (including nanostars) may allow for the intersection of emitted electrons with other surfaces of the emitter geometry. These effects are not presently accounted for, although they should be negligible for the tip-like emission described in the present work.”

6. Bainbridge and Bryan (NJP, 2014) have reported on velocity map imaging of photoelectrons emitted from nanotips and Park et al. (PRL, 2012) have reported on strong-field effects in the angular distribution of photoemitted electrons from metal tips. Those papers might be cited.

We appreciate the suggestion and these references have now been added to the penultimate Introduction paragraph on p. 4: “Angular photocurrent mapping and steering have also been demonstrated for gold²⁹ and tungsten^{30,31} nanotips, primarily in the field emission regime.”

Reviewer #2 (Remarks to the Author):

Referee report on article NCOMMS-19-32909

Plasmonic Nanostar Photocathodes for Optically-Controlled Directional Currents

by Pettine, Nesbitt et al.

The paper NCOMMS-19-32909 addresses a topic of high current interest, namely the optical control over directional currents with the potential of fs time resolution. The authors bring in their excellent expertise in the field of nanoscale plasmonic resonators and emitters. The paper is well written and scientifically sound. The data are of high interest to a wide readership in the field of ultrafast spectroscopy, diffraction and microscopy, terahertz technology and even accelerator physics.

Two major and some minor points of criticism are listed below.

I recommend publication in Nature Communications after the authors have met my major points and addressed or commented on the minor points.

Major points:

1. The authors discuss the intensity dependence for a specific star as follows:

Log-log intensity dependence plots and power-law fits are shown for the three sample nanostars, with the Star 2 resonance ($n = 3.6 \pm 0.1$) representing an intermediate process with weighted contributions from both 3PPE and 4PPE.

If indeed weighted contributions from 3PPE and 4PPE occur for Star 2, it must be considered that the two contributions have a different power dependence. Mathematically, the logarithm of a sum of two power terms (Eq. 1) does not result in a linear dependence in the double-logarithmic plot. The authors should address this mathematical argument, re-examine these results and explain the linear dependence shown in Fig. 1e, star 2 more carefully. Maybe one of the contributions is negligible so that the deviation from linear is very small? But that would not fit to $n=3.6$

To me, something seems wrong here.

We appreciate the opportunity to clarify this point and to this end have added a much more careful discussion (including new Eq. 2) on p. 7-8 regarding the determination of effective nonlinear process orders via the intensity-dependence slope on a log-log plot. We additionally note in the caption for Fig. 1e that “Intermediate values between 3 and 4 are effective process orders representing a weighted average of 3PPE and 4PPE contributions.”

Over larger intensity ranges than studies here, as the slope varies more clearly between $n = 3$ to $n = 4$, a single effective-process-order fit will become quite poor but still yield an intermediate process order. However, as demonstrated in Fig. 1e the fit remains quite good for the intensity range investigated. This simple manner of defining effective process order or effective nonlinearity has also been used, e.g., by Yalunin & Ropers, *Phys. Rev. B* (2011) and by Lehr & Elmers, *JPC C* (2019). We hasten to add that the primary role of the intermediate process order information is to indicate that such a transition does occur and to indicate where it occurs in the present studies, but the precise transitional behavior will vary with particle work function (and thus depends on material, geometry, local environment), overall laser intensity, laser linewidth, etc. By contrast, the demonstration of shifting between a clear $n = 3$ regime to a clear $n = 4$ regime is of critical importance to demonstrating that the emission is predominantly MPPE rather than strong-field or thermionic emission.

2. Is the dI/dE dependence shown in Figure 2h really compatible with a 3PPE process? I would have expected a differently-shaped function. For the derivative one would expect a negative value on the falling wing, because the intensity decreases with increasing kinetic energy. Maybe it does not show the derivative but simply the spectrum? Moreover, I would find it very helpful to assist the reader with a few sentences about the role of the ponderomotive forces and how much they actually contribute to the kinetic energy in Figure 2h. This must be strongly depend on the incident light intensity, a fact that should be stated in the text.

We thank the reviewer for pointing out this opportunity to clarify the notation. Figure 2h shows the photoelectron kinetic energy spectrum for the described 4PPE process, not the derivative thereof. To avoid confusing and unnecessary notation, we have changed the vertical axis labels for both Fig. 2g and 2h to “Normalized Counts”. In terms of the shape of this 4PPE spectrum, we note that the photocurrent is pulled down to zero at the origin by a p_z weight factor with respect to the surface normal during emission (as derived in the Methods), which ultimately becomes a factor of \sqrt{E} , in addition to a \sqrt{E} free-space density of states factor.

To clarify the role of the ponderomotive acceleration and make the intensity dependence more apparent, we have added a discussion of these points on p. 12.

Minor points:

3. The wet-chemical growth and subsequent dialysis might leave residues of solvent or sucrose on the surface of the nanostars. An ultrathin insulating (mono-)molecular layer between one of the tips of the nanostar and the conductive substrate might give rise to gap resonances (as discussed in Phys. Rev. Lett. 108 (2012) 237602). This would explain the strength of emission and the high finesse of the nanostar resonators. It might also explain why the tips pointing away from the surface obviously do not emit (at least not significantly). Since some of these are strongly tilted, the photon polarization argument alone cannot explain this. Due to irregularities in the shape, such gap resonances could possibly also occur even without an insulating spacer layer.

Although to some extent speculative, I feel that the authors should at least mention the possible presence of gap resonances.

This is a valuable point, for which we have added a discussion of gap resonances on p. 12-13. While the observed nanostar MPPE behaviors can be accounted for in the context of the sharp nanostar tips, in-plane polarization, and highly nonlinear polarization dependence – as demonstrated in the Figs. 2-5 and Supplementary Fig. 2 field enhancement/photocurrent simulations – this does not preclude possible contributions from gap resonances. To this end we have also added a new panel (c) to Supplementary Fig. 2 to display the field enhancements for a clearer view of the region beneath the nanostar where such gap resonances may occur, although the presence of a gap resonance will depend on the specific configuration of this nanostar-HEPES-ITO interface, which we do not have direct access to.

4. In the context of femtosecond emitters the dephasing time (T_2) certainly plays a role. Could the authors please comment on how T_2 enters into the controlled emission process. The “ringing plasmon” might pose a limitation on the achievable time structure. Or maybe I overlook something here?

The plasmon dephasing (T_2) time indeed represents a fundamental temporal limitation on photocurrent control in the systems described here. On this point, we highlight two locations in the manuscript:

- (1) On p. 5, we briefly note that “Photocurrent control timescales approaching the attosecond range may be achievable, fundamentally limited only by the nonlinear photoemission decay associated with plasmon dephasing.”
- (2) On p. 19-20, we explain in more detail that “The photoemission switching timescale is fundamentally limited by plasmon dephasing and by the optical cycle of the excitation laser field, which defines the fastest timescale on which polarization and frequency can be manipulated. Due to the E^{2n} process nonlinearity, the n PPE current decays $2n$ -times faster than the plasmonic field, which typically dephases in 10 fs or less^{29,42}. Thus, the 3PPE or 4PPE decay times are comparable to the 1-3 fs optical cycles for visible and near-infrared frequencies. This suggests that spatiotemporal control over hot spot excitation and directional current generation may be achieved on timescales approaching the attosecond range, even in the weak-field MPPE intensity regime.”

To clarify this further, we have changed the phrasing from “fundamentally limited by plasmon dephasing and ...” to “fundamentally limited by the plasmon dephasing (T_2) time and ...”.

5. The authors might want to rephrase a sentence in the abstract: In the nano-world, R 3.4 nm is sharp but not “point like”.

While precedent exists in the nanotip literature for the term “point-like” for even larger sources than discussed here (e.g. Ropers & Elsaesser, *PRL* (2007); Mueller & Ernstorfer, *ACS Photonics* (2016)), this is generally with regard to the spatial coherence characteristics and/or by contrast with extended sources. In the present case, we recognize the reviewer’s argument and have replaced “point-like tips” with “sharp tips ($R_{\text{tip}} = 3.4 \text{ nm}$)” in the abstract.

6. My last point concerns the way, how a closely-related previous publication is mentioned.

The abstract states: “However, angular photocurrent distributions in nanoplasmonic systems remain poorly understood...”

And on p.4: “Such capabilities have only appeared recently (24,26-28)...”

I feel that the results of ref. 27 (*Nano Lett.* 17, 6606-6612 (2017)) should be mentioned in some more detail as in the short remark on p.4. That paper deals with the closely-related issue of plasmonic emission from Au nanorods on an ITO substrate, with backside illumination with fs-pulses from a Ti:sapphire laser using a momentum microscope that also resolves individual nanorods in real-space imaging mode. This was actually the first study of the momentum- and energy-distribution of plasmonic electron emission from resonantly-excited, individual, selected Au nanorods. The directional emission from the tips (with the rods lying flat on the surface, identical to the emitting branches of the nanostars) has been resolved not only on a true momentum scale (calibrated in inverse Angstroms) but also on the energy scale, giving valuable additional information on the emission process.

The present work on nanostars is excellent and the paper is very convincing. However, the fact that a gold nanotip pointing parallel to the surface is a highly efficient plasmonic emitter has been shown in ref. 27. I list this under the minor points and leave it to the authors how far they implement a fair judgement of the ref. 27 work.

We agree that the important work performed in ref. 27 should be addressed further. To this end, we have added the following description to p. 4:

“Full photoelectron momentum and energy characterization has been achieved by Lehr *et al.* on individual gold nanorods and bow-tie nanoantennas using time-of-flight momentum PEEM^{27,28}, which serves to clarify nanoplasmonic angular photoemission distributions and phenomena such as the transition into the optical field emission regime.”

Reviewers' Comments:

Reviewer #1:

Remarks to the Author:

The authors have responded in detail and convincingly to the questions and comments of both Reviewers. I now gladly recommend to publish these nice results in Nature Communications.

Reviewer #2:

Remarks to the Author:

The authors adequately addressed all my points of criticism. I have no further comments, the paper is ready for publication.

Reviewer #1 (Remarks to the Author)

The authors have responded in detail and convincingly to the questions and comments of both Reviewers. I now gladly recommend to publish these nice results in Nature Communications.

Christoph Lienau

Reviewer #2 (Remarks to the Author)

The authors adequately addressed all my points of criticism. I have no further comments, the paper is ready for publication.

We sincerely thank both reviewers for their insightful comments and critique.